# Sustaining Retirement during Lockdown: Annuitized Income and Older American's Financial Well-Being before and during the COVID-19 Pandemic

**Qi Sun [1,*] and Gary Curnutt [2]**

[1]  Pacific Life Insurance Company, Newport Beach, CA 92660, USA
[2]  The School of Accounting, Finance, Information Systems and Business Law, College of Business, Western Carolina University, Cullowhee, NC 28723, USA; gcurnutt@wcu.edu
[*]  Correspondence: qi.sun@pacificlife.com

**Abstract:** The landscape of employer-sponsored retirement plans in the U.S. has changed dramatically during the past few decades as more and more private-sector employers have decided to freeze or terminate traditional pension plans. Defined contribution (DC) plans became the primary choice or the only choice for employees to participate in employer-sponsored retirement plans. In the next ten to twenty years, the income from pension plans will only count for a third of the total retirement income for GenXers when compared to their baby boomer counterparts. It is important for research to provide evidence on how the change in retirement income resources impacts retirees' retirement security and financial wellness. Using Health and Retirement Study (HRS) data before and during the COVID-19 pandemic, this study examines the association between annuitized income and various measures of older Americans' financial well-being over time, particularly during the pandemic. This study finds that receiving annuitized income has a statistically significant relationship with reduced subjective financial well-being for both measurements, while only one of the measures of objective well-being, having liquid assets greater than the median household income, has a statistically significant positive relationship with receiving annuitized income.

**Keywords:** financial well-being; longevity risk; annuitized income; defined contribution plans

## 1. Introduction

The goal of this paper is to examine the association between annuitized income and various measures of older Americans' financial well-being. In this paper, we use the Health and Retirement Study (HRS) data before and during the COVID-19 pandemic. This allows us to gauge the relationship and observe the effect over time, particularly during the pandemic. Four separate logistic models are used to evaluate the relationship between measures of both objective and subjective financial well-being and annuitized income, as well as other control variables, during the COVID-19 pandemic. We find statistically significant negative relationships for both our measures of subjective financial well-being and receiving annuitized income. We find only one of our measures of objective well-being, having liquid assets greater than the median household income, has a statistically significant positive relationship with receiving annuitized income.

Past studies have described retirement income in the U.S. as a "three-legged stool", including annuitized income from social security, employer-sponsored pensions, and withdrawals from personal savings (Brown 2009; Collins and Yeskel 2011). Among these three income resources, the "social security" leg and "pension" leg serve as the collective risk, pooling risk across all beneficiaries and providing guaranteed lifetime income. The "personal savings" leg is better characterized as individualized risk but only represents one-third of the structure (Herd 2009). However, as employer-sponsored retirement plans shift from pension plans, or defined benefit (DB) plans to defined contribution (DC) plans,

this "three-legged stool" has transformed into a "two-legged stool", with an increased portion in personal savings and DC plans. Under the "two-legged stool" retirement income structure, social security represents the majority of indexed guaranteed lifetime income and pooled collective risk. However, social security income is limited and only acts as a supplemental source of income (Hervani et al. 2020), which leaves future retirees with a retirement income portfolio that has a lower annuitized lifetime income element and concentrates the portfolio on income with individualized risk.

Having a portion of annuitized income not only helps retirees diversify their individualized retirement income risk but also enhances their financial well-being. Chatterjee et al. (2011) discovered that holding a DB plan positively impacts retirees' financial confidence. Bender (2012) concluded that the presence of a DB plan would mitigate the negative effect of personal risk faced by DC plan participants on job satisfaction (Bender 2012). Similarly, Lim and Lee (2021) found that the positive effect of DB plans on subjective financial well-being remains as individuals age. However, the positive effect of DC plans and IRAs diminishes when individuals approach retirement age. From the income stream perspective, Panis (2004) pointed out that those who heavily rely on social security income to finance their retirement are less likely to feel satisfied with their retirement life (Panis 2004). Conversely, those who receive lifetime annuitized income from DB plans are more likely to feel more satisfied.

This paper is theoretically grounded on the cumulative inequality theory (CI theory). Early research on the CI theory started by linking the cumulative advantage theory with the life course theory (Crystal and Shea 1990; Dannefer 1987). The progressive nature of the CI theory naturally lent itself to gerontological research to explore later-life poverty and disadvantages (Crystal 2018; O'Rand 1996). Ferraro and Shippee (2009) established five axioms of the CI theory that can be used to identify how life course trajectories are influenced by early and accumulated inequalities and how these can be modified by access to resources. The second axiom of the CI theory explains that disadvantages should be treated as an exposure to risk instead of negative outcomes since people with disadvantages need to solve this situation by encountering more potential risks than those who do not have such disadvantages (Castro Baker et al. 2019; Ferraro and Shippee 2009). The CI theory also points out that inequality may diffuse across life domains (DiPrete and Eirich 2006). For example, the disadvantage in wealth can spill over to the mental health domain and cause disadvantages in stress and depression. Under the CI theory, we consider not receiving annuitized income from DB plans/annuity to be a disadvantage because of the individualized income risk from the "two-legged stool" retirement income structure; such a disadvantage will expose households to more potential risks and further spill over to the domain of their financial wellness. This paper provides empirical evidence on how receiving annuitized income from DB plans/annuities impacts objective and subjective financial well-being separately. Unlike previous literature that either focuses on individuals' subjective perceptions of life/retirement satisfaction or their objective financial capacities, such as income and consumption, this study used four measurements to proxy both subjective and objective financial well-being and tested whether annuitized income has a consistent impact on both of them. Additionally, this study picked two continuous waves of data before and during the COVID-19 pandemic and checked whether the impact of annuitized income on financial well-being varied because of the global pandemic. Lastly, we examined the interaction effect between annuitized income and mental/physical health on financial well-being. Past studies found evidence that physical and mental health conditions are directly related to individuals' financial well-being (e.g., Çelik et al. 2017; Branch-Allen and Jayachandran 2016; Temple and Williams 2018). This study contributes to this body of literature by exploring whether annuitized income mediates the negative effect of poor mental/physical health on individuals' financial well-being. The research findings provide insights for financial service professionals, educators, and policymakers to assist baby boomers and Generation X, who have a lower annuitization rate due to the shift from DB plans to DC plans, as well as younger generations who may have no annuitized income

from employer-sponsored retirement plans. The positive effect of annuitized income on subjective well-being cannot be neglected, especially for older retirees who face a decline in both health conditions and cognitive abilities. This "hand-off" lifetime income stream needs to be brought back to the DC plan to enhance participants' retirement income security and financial well-being.

## 2. Materials and Methods

This paper uses data from the Health and Retirement Study (HRS). The HRS is a longitudinal panel conducted by the University of Michigan. Both the National Institute on Aging (NIA) and the Social Security Administration support the HRS. When weighted, the HRS is a nationally representative sample of about 20,000 respondents, collected in the U.S. This paper uses panel data from the wave in 2016 and the wave in 2020. Four separate logistic models are used to evaluate the relationship between measures of both objective and subjective financial well-being and annuitized income, as well as other control variables, during the COVID-19 pandemic and recession. Table 1 shows the descriptive statistics of our analysis sample.

**Table 1.** Descriptive statistics of analysis sample.

| Variable | Mean (Standard Error) | |
|---|---|---|
| | 2016 * | 2020 |
| Debt to Asset Ratio < 0.5 (Yes: 1) | 0.8170 (0.0088) | 0.8437 (0.0072) |
| Total Liquid Assets > Median Household Income (Yes: 1) | 0.2315 (0.0097) | 0.2880 (0.0090) |
| Having Difficulties Paying Bills (Yes: 1) | 0.5671 (0.0112) | 0.4732 (0.0099) |
| Being Upset about Ongoing Financial Strain (Yes: 1) | 0.2425 (0.0098) | 0.1484 (0.0071) |
| Receiving Annuitized Income (Yes: 1) | 0.1912 (0.0081) | 0.2459 (0.0086) |
| Age | 66.2087 (0.2153) | 69.8703 (0.2048) |
| Female | 0.5170 (0.0114) | 0.5696 (0.0099) |
| Race | | |
| White | 0.8583 (0.0070) | 0.7564 (0.0086) |
| Black | 0.0801 (0.0048) | 0.1650 (0.0074) |
| Other | 0.0616 (0.0054) | 0.0785 (0.0054) |
| Years of Education | 13.8679 (0.0562) | 13.7045 (0.0531) |
| Married | 0.5292 (0.0113) | 0.4645 (0.0099) |
| Self-Reported Health | | |
| Excellent | 0.0889 (0.0067) | 0.0797 (0.0054) |
| Very Good | 0.3474 (0.0111) | 0.3300 (0.0094) |
| Good | 0.3544 (0.0107) | 0.3653 (0.0096) |
| Fair | 0.1668 (0.0080) | 0.1845 (0.0077) |
| Poor | 0.0426 (0.0043) | 0.0405 (0.0039) |
| Depression Score | 1.2140 (0.0405) | 1.3070 (0.0380) |
| Total Annual Household Income (in USD) | 94,507.49 (3473.3466) | 74,279.03 (2976.2586) |

**Table 1.** *Cont*.

| Variable | Mean (Standard Error) | |
|---|---|---|
| | **2016 *** | **2020** |
| Financial Wealth (in USD) | 291,721.79 (22,584.0423) | 351,957.57 (47,714.7239) |
| Housing Wealth (in USD) | 243,411.45 (8210.8105) | 265,318.42 (6881.0792) |

Note: Observations from 2016 and 2020 HRS Data. * Weighted mean for wave 2016. Wave 2020 is unweighted due to unavailable individual weight variable data.

The key explanatory variable is owning annuitized income. This is a binary variable that pulls from two questions on the survey, a question that asks about pension ownership and another that asks about the income that comes from annuities. If the respondent or their spouse indicated that they have either a pension or income from annuities, the key explanatory variable, annuitized income, is coded as a 1. Otherwise, the key explanatory variable is coded a 0.

The subjective measures of financial well-being come from the leave-behind survey. This survey is a subsample of the HRS; however, the respondents are alternated each wave. Starting in 2006, this subsample included one-half of the entire HRS sample, selected at random. Those who are selected to participate in the leave-behind survey are given the opportunity to complete a self-administered Psychology and Lifestyle Questionnaire and an interview. The Psychology and Lifestyle Questionnaire is used to obtain information about the participants' evaluations of their life circumstances, subjective well-being, and lifestyle. Due to the rotation of the subsample in each 2-year wave, longitudinal data using the same respondents are only available at four-year intervals. For this reason, we used 2016 and 2020 data to observe the same individuals before and during the COVID-19 recession.

The two subjective measurements of financial well-being are financial strain and difficulties paying bills. The financial strain variable is a binary variable that uses the question asking if the respondent was upset due to ongoing financial strain. We coded the variable as 1 if the individual responded that they have an ongoing financial strain, and it is either somewhat upsetting or very upsetting. The variable is coded as 0 if the respondent reported no ongoing strain or some strain but it is not upsetting. In a similar fashion, the difficulties paying bills variable is also a binary variable. The variable is coded as a 1 if the respondent reported any difficulties paying bills, including not very difficult, very difficult, and completely difficult. If the respondent answered "not at all difficult" the variable is coded as a 0.

The two objective measures that serve as dependent variables are a debt-to-asset ratio of less than 0.5 and having liquid assets greater than the median household income. Previous studies on debt and the financial well-being of older workers and retirees use a 0.5 debt-to-asset ratio as a threshold to gauge financial vulnerability and make comparisons across age groups and cohorts entering retirement (Lusardi et al. 2020). Although interpreting financial ratios can be nuanced, existing literature suggests that having a debt-to-asset ratio of less than 0.5 suggests acceptable levels of debt (Winger and Frasca 2006). Having a debt-to-asset ratio higher than this is associated with greater difficulty repaying debt obligations. Furthermore, the debt-to-asset ratio should decrease with age (Keown 2015), implying a ratio of 0.5 or greater should indicate a position of financial vulnerability for older individuals. Having liquid assets greater than the median gross annual household income is likewise a binary variable; households who report liquid assets greater than the median are coded as a 1, while households who do not are coded as a 0. Previous studies have compared the value of liquid assets to household income as a means to gauge financial vulnerability as it could indicate how well a household may cope with a sudden or unexpected expense (Lusardi et al. 2020).

This paper estimates four random-effects (REs) logistic models. For each model, the individual's financial well-being is the dependent variable and there are two models

with subjective financial well-being (being upset because of ongoing financial strain and difficulties paying bills) and two models with objective financial well-being (debts-to-assets ratio and liquidity). The models are further parsed by the key explanatory variable, receiving some form of annuitized income from defined benefit plans or annuity products. Control variables include age, gender, marital status, race, years of education, self-reported health status, depression score, log-transferred annual household income, financial wealth, and housing wealth.

The result is four separate logistic models:

$$Y_{it}{}^{S*} = \alpha_0 + \alpha_1 AI_{it} + \alpha_k X + \nu_i + \theta_{it}$$
$$Y_{it}{}^{B*} = \beta_0 + \beta_1 AI_{it} + \beta_k X + \zeta_i + \varepsilon_{it}$$
$$Y_{it}{}^{D*} = \gamma_0 + \gamma_1 AI_{it} + \gamma_k X + \eta_i + \lambda_{it}$$
$$Y_{it}{}^{L*} = \delta_0 + \delta_1 AI_{it} + \delta_k X + \tau_i + \mu_{it}$$

$$Y_{it}{}^{S} = \begin{cases} 1, & Y_{it}{}^{S*} > 0 \\ 0, & Y_{it}{}^{S*} \le 0 \end{cases}$$

$$Y_{it}{}^{B} = \begin{cases} 1, & Y_{it}{}^{B*} > 0 \\ 0, & Y_{it}{}^{B*} \le 0 \end{cases}$$

$$Y_{it}{}^{D} = \begin{cases} 1, & Y_{it}{}^{D*} > 0 \\ 0, & Y_{it}{}^{D*} \le 0 \end{cases}$$

$$Y_{it}{}^{L} = \begin{cases} 1, & Y_{it}{}^{L*} > 0 \\ 0, & Y_{it}{}^{L*} \le 0 \end{cases}$$

$$\nu_i \sim N\left(0, \sigma_\nu{}^2\right)$$
$$\zeta_i \sim N\left(0, \sigma_\zeta{}^2\right)$$
$$\eta_i \sim N\left(0, \sigma_\eta{}^2\right)$$
$$\tau_i \sim N\left(0, \sigma_\tau{}^2\right)$$

$$\theta_{it}, \varepsilon_{it}, \lambda_{it}, \mu_{it} \sim N\left(0, \frac{\pi^2}{3}\right)$$
$$i = 1, 2, \ldots, I$$
$$t = 1, 2, \ldots, T$$
$$k = 2, 3, \ldots, K$$

where the latent variable $Y_{it}{}^{*}$ is the unobserved net benefit of financial well-being for individual $i$ at time $t$; the variable $Y_{it}{}^{S}$ is the observed dichotomous decision of individual $i$ reporting ongoing financial strain at time $t$; the variable $Y_{it}{}^{B}$ is the observed dichotomous decision of individual $i$ reporting difficulties paying bills at time $t$; the variable $Y_{it}{}^{D}$ is the observed dichotomous decision of individual $i$ reporting a debts-to-assets ratio less than 0.5 at time $t$; and the variable $Y_{it}{}^{L}$ is the observed dichotomous decision of individual $i$ reporting having total liquid assets > household median annual income at time $t$.

$AI_{it}$ represents receiving annuitized income for individual $i$ at time $t$ and $X$ is a matrix of other explanatory variables which include age, gender, marital status, race, years of education, self-reported health status, depression score, log-transferred annual household income, financial wealth, and housing wealth.

$\alpha_0 / \beta_0 / \gamma_0 / \delta_0$ is the intercept and $\alpha_k / \beta_k / \gamma_k / \delta_k$ is the $k$th regression coefficient, measuring the effect of one unit change of independent variables on the observed dichotomous dependent variables. $\nu_i / \zeta_i / \eta_i / \tau_i$ is the unobserved effect which is assumed to be unrelated to each explanatory variable and time-varying error in all time periods, and is distributed with mean 0, and variance $\sigma_\nu{}^2$, $\sigma_\zeta{}^2$, $\sigma_\eta{}^2$, $\sigma_\tau{}^2$. The error $\theta_{it} / \varepsilon_{it} / \lambda_{it} / \mu_{it}$ is the time-varying error which represents unobserved factors that change over time and affect the latent dependent variable, and is distributed with mean 0 and the variance $\frac{\pi^2}{3}$, a standard logistic density, stated in Bosker and Snijders (2011) and Li et al. (2011).

This study used the REs model for several reasons. According to Allison (2009), the fixed-effects (FEs) model needs variability within the subject to function well. If there is little variability within objects across time, the standard error from the FEs model will be too large to accept. In our sample group, both the subjective and objective financial well-being measurements did not change too much between the years 2016 and 2020 and we saw that this affected the standard errors estimated from FEs model results in the sensitivity test session (Appendix A). Second, this paper focuses on whether the group who does not have annuitized income is more disadvantaged in financial well-being, compared to the other group. This is a between-group comparison but not a within-group comparison across time. This question will be answered more appropriately by REs but not FEs models. For the above reasons, both Wooldridge (2010) and Allison (2009) stated that REs models are still desired/preferred under such circumstances. In conclusion, even though we are aware that, besides control variables, there are unobserved variables correlated with explanatory variables and these results omitted variable bias, we decided to go with REs models in the main session and to present the FEs model results in the sensitivity session as an additional support.

In conclusion, this paper tests the below hypotheses:

**H1.** *Receiving annuitized income from DB plans/annuities will be positively associated with objective and subjective financial well-being.*

**H2.** *There will be a difference in the positive relationship before and during the pandemic.*

**H3.** *Annuitized income will moderate the relationship between mental/physical health and objective and subjective financial well-being.*

## 3. Results

This paper provides empirical evidence on how receiving annuitized income from DB plans/annuities impacts objective and subjective financial well-being separately. However, our results are mixed. Table 2 summarizes the estimated odds ratio from each of the random effects logistic regression models that measure the subjective or objective financial well-being of our explanatory and control variables.

### 3.1. Annuitized Income and Subjective Measures of Financial Well-Being

Our results show that receiving some forms of annuitized income from DB plans or annuity products is associated with a decrease in the odds of having troubles with subjective financial well-being (statistically significant in both measurements). Individuals who receive some form of annuitized income have reduced odds, odds ratio of 0.60, or about 40 percent reduced odds of reporting financial strain, compared to those who have no form of annuitized income. Likewise, individuals receiving annuitized income have an odds ratio of 0.54, or about 46 percent reduced odds of reporting difficulty paying bills, compared to those who have no form of annuitized income. The findings indicate that receiving annuitized income significantly relieved people's anxiety about financial issues. This positive effect on relieving financial strain stresses still holds after running a fixed effects logistic model as a sensitive test (results are shown in Appendix A).

Our results show that the pandemic year of 2020 reduced the odds of reporting both financial strain and difficulty paying bills for our analysis sample. The 2020 pandemic brought about much economic uncertainty and resulted in an overall reduction in labor. We offer several contributory factors that may help explain the reduction in odds for subjective financial difficulties for our analysis sample, despite the objective economic downturn. With our analysis sample being squarely within the retirement range, average age being about 66 in wave 2016 and about 69 in wave 2020, perhaps the participants in our sample simply started to retire. Thus, individuals newly entering retirement may have a brighter subjective outlook on their financial well-being.

**Table 2.** Odds ratios for objective and subjective measures of financial well-being.

| Independent Variable | Odds Ratio (Standard Error) | | | |
|---|---|---|---|---|
| | Financial Strain | Difficulty Paying Bills | Debt-to-Asset < 0.5 | Liquid Assets > Median HH Income |
| Receiving Annuitized Income | 0.5993 ** | 0.5430 *** | 1.0362 | 1.3183 * |
| | (0.1075) | (0.0841) | (0.1298) | (0.1503) |
| Pandemic Year (2020) | 0.4355 *** | 0.5124 *** | 1.1213 | 1.9377 *** |
| | (0.0498) | (0.0521) | (0.0683) | (0.1483) |
| Annuitized Income # Pandemic Year | 1.0415 | 1.2068 | 1.1840 | 0.7615 |
| | (0.2846) | (0.2508) | (0.2029) | (0.1117) |
| Age | 0.9468 *** | 0.9513 *** | 1.0994 *** | 1.0363 *** |
| | (0.0059) | (0.0055) | (0.0048) | (0.0043) |
| Married | 0.6110 *** | 0.7447 ** | 0.8975 | 0.8366 * |
| | (0.0728) | (0.0850) | (0.0685) | (0.0725) |
| Gender Reference group is Male | 1.3901 ** | 1.2451 | 0.8173 ** | 1.1352 |
| | (0.1646) | (0.1413) | (0.0606) | (0.0952) |
| Years of Education | 1.0101 | 0.9404 ** | 0.9028 *** | 0.9692 |
| | (0.0204) | (0.0192) | (0.0122) | (0.0164) |
| Race—Black Reference group is White | 0.6494 ** | 1.0312 | 0.7538 ** | 0.7525 * |
| | (0.0935) | (0.1515) | (0.0664) | (0.1032) |
| Race—Other Reference group is White | 0.6801 | 1.1990 | 1.0008 | 0.9504 |
| | (0.1380) | (0.2518) | (0.1164) | (0.1509) |
| Liquid Assets > Median HH Income | 0.1776 *** | 0.1977 *** | | |
| | (0.0329) | (0.0259) | | |
| Debt-to-Asset < 0.5 | 0.3465 *** | 0.3042 *** | | |
| | (0.0441) | (0.0435) | | |
| Depression | 1.3596 *** | 1.2253 *** | 0.9321 *** | 1.0115 |
| | (0.0402) | (0.0377) | (0.0161) | (0.0228) |
| Health Reference group is Excellent | | | | |
| Very Good | 1.1820 | 1.7983 ** | 0.6483 ** | 1.1572 |
| | (0.2776) | (0.3425) | (0.0876) | (0.1518) |
| Good | 2.1619 *** | 3.3828 *** | 0.6056 *** | 1.1495 |
| | (0.5021) | (0.6608) | (0.0822) | (0.1562) |
| Fair | 3.5794 *** | 5.2486 *** | 0.5787 *** | 0.9627 |
| | (0.8892) | (1.1606) | (0.0850) | (0.1546) |
| Poor | 4.1178 *** | 4.1082 *** | 0.4560 *** | 0.9832 |
| | (1.2891) | (1.2708) | (0.0861) | (0.2517) |
| Log of Annual Household Income | | | 0.9284 *** | 1.0766 *** |
| | | | (0.0089) | (0.0173) |
| Log of Non-Housing Net Worth | | | 1.1265 *** | 3.2009 *** |
| | | | (0.0079) | (0.1236) |
| Log of Housing Net Worth | | | 1.1121 *** | 1.0039 |
| | | | (0.0064) | (0.0072) |
| lnsig2u | 3.2158 *** | 4.7296 *** | 4.5555 *** | 3.8532 *** |
| | (0.5578) | (0.5894) | (0.3546) | (0.3564) |
| N | 6020 | 6041 | 18,575 | 24,421 |

Note: Observations from 2016 and 2020 HRS Data. *** Denotes statistical significance at the 1% level, ** denotes statistical significance at the 5% level, and * denotes statistical significance at the 10% level.

During this time frame, the average net worth also increased for these individuals. Although this is not abnormal for a longitudinal study, an interesting point of future research could be to more deeply explore the impact of the pandemic recession on the standard life-cycle wealth accumulation leading into retirement. From an analysis of the descriptive statistics of our analysis sample, perhaps the negative economic impact of the pandemic recession had a lesser impact on those entering retirement than those still working. In addition, several governmental financial "Safety-nets" were created or expanded. These governmental initiatives could play a role in dampening the effect of the recession on subjective financial well-being; this too could be a very interesting avenue of future research.

As a proxy for mental health, the depression score also had a statistically significant relationship with both measures of subjective financial well-being in our analysis sample. The odds ratio for the depression score and reporting financial strain was 1.36, or about a

36 percent increase in the odds of reporting financial strain for every incremental increase in depression score.

Subjective, i.e., self-reported, health status also had a statistically significant relationship with subjective financial well-being. In general, the poorer the individual's self-reported health status, the greater the odds of reporting subjective financial difficulties. For reporting financial strain, compared to those with excellent health, those who report being in good health have an odds ratio of 2.16, a 116 percent increase in the odds of reporting financial strain. For those who self-report as being in fair health, their odds ratio is 3.58, a 257 percent increase in the odds of reporting financial strain. Finally, those who report being in poor health have 312 percent higher odds of reporting financial strain than those in excellent health.

### 3.2. Annuitized Income and Objective Measures of Financial Well-Being

In terms of objective financial well-being, we only found a marginally significant positive effect of annuitized income on the liquidity measurement. With an odds ratio of 1.32, the marginal relationship suggests that receiving annuitized income results in about a 32 percent increase in the odds of having liquid assets greater than the median household wealth.

Likewise, the pandemic year only had a statistically significant relationship with one measure of objective financial well-being. For the pandemic year, individuals in our sample had about 94 percent, odds ratio of 1.94, higher odds of having liquid assets greater than the median household income. As described above, the net worth of our analysis sample did increase as they entered normal retirement years. This shift into retirement could play a role in explaining this relationship as retirees started to take distributions from DC plans. Another vein of future research could be to explore the change in liquid assets as individuals enter retirement and seek to explore if the pandemic attributed to any additional liquid withholdings, as our study was not designed to specifically parse this relationship.

The depression scores only had a statistically significant relationship with the debt-to-asset ratio less than 0.5 variable. With an odds ratio of 0.93 for each incremental increase in depression score, there is about a 7 percent decrease in the probability of having a debt-to-asset ratio of less than 0.5.

Physical, self-reported health is also related to the objective financial measure of a debt-to-asset ratio less than 0.5. Overall, the poorer the self-reported health status, the lower the odds of having less debt. Compared to those in excellent health, those in very good health, with an odds ratio of 0.65, have about a 35 percent decrease in the odds of having a debt-to-asset ratio of less than 0.5. Likewise, those in good health, with an odds ratio of 0.61, in fair health, with an odds ratio of 0.58, and in poor health, with an odds ratio of 0.46, have decreased odds of having a debt-to-asset ratio of 0.39 percent, 0.42 percent, and 0.34 percent, respectively.

### 3.3. Predicted Probabilities and the Moderating Effect of Annuitized Income

To further analyze our data, in light of our hypothesis, we predicted the probabilities of subjective and objective financial well-being measures over the depression score and health status for those receiving/not receiving annuitized income to show the moderating effect of annuitized income between several variables.

Figure 1 shows the predicted probability of being upset about financial strain across depression scores. The solid line shows the predicted level of being upset about financial strain given a level of depression score without receiving annuitized income. The dashed line shows the predicted probability of being upset about financial strain given a level of depression score while receiving annuitized income. The figure shows that those who have annuitized income have lower predicted probabilities of being upset about financial strain across all levels of depression score. Additionally, the disparity between the two gradually increases as depression score increases. For example, when the depression score

is 0, the predicted probability for those who receive annuitized income is 4% lower than those who do not receive annuitized income, holding other variables at means, while the difference in probability increased to 7% when the depression score was 8. Even though the extent is economically small, it is statistically significant. Further research into the effects of annuitized income on mental health, and subjective measures of financial well-being could deeper explore this relationship.

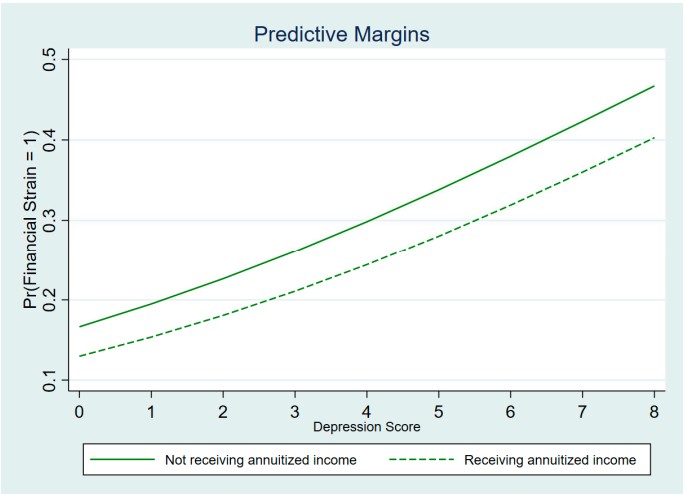

**Figure 1.** Predicted probability of being upset about financial strain across depression scores, with/without annuitized income.

Annuitized income could also have a moderating effect on the relationship between subjective financial well-being and subjective physical well-being. Figure 2 shows the predicted probability of being upset about financial strain given varying levels of self-reported health status. Similar to Figure 1, Figure 2 shows that those who have annuitized income have a lower probability of being upset about financial strain than those who do not. Figure 2 also shows an increasing disparity between the two as health status decreases.

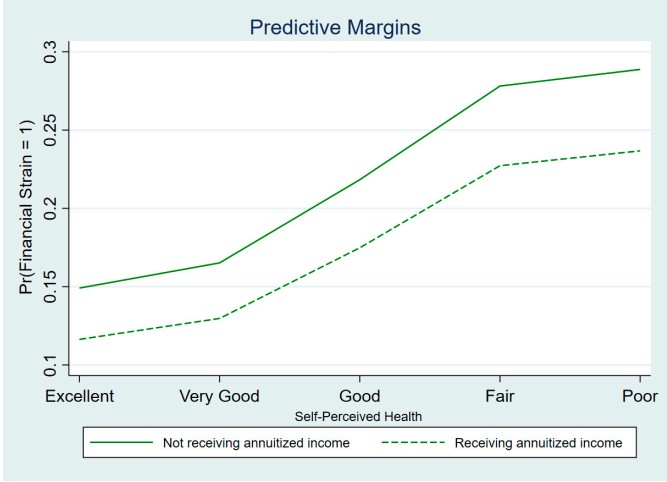

**Figure 2.** Predicted probability of being upset about financial strain across health conditions, with/without annuitized income.

Figure 3 depicts the moderating effect of annuitized income between having a debt-to-asset ratio of less than 0.5 given varying levels of reported depression score. Individuals who have an annuitized income have a higher predicted probability of having a debt-to-asset ratio of less than 0.5 than those who do not have an annuitized income, at every level of depression score. However, the two lines are considerably parallel.

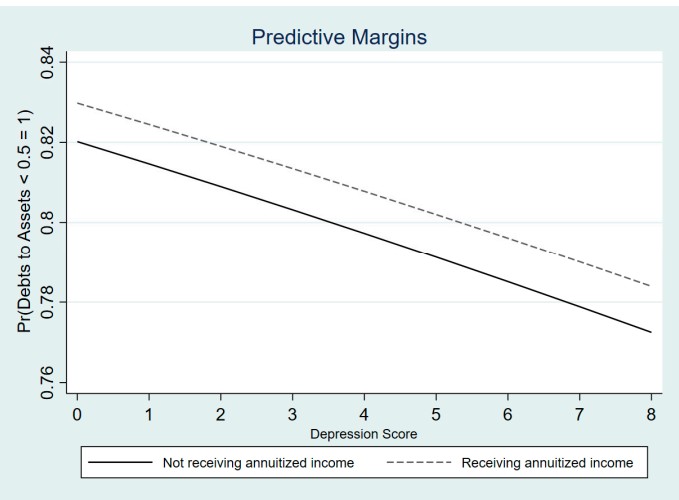

**Figure 3.** Predicted probability of having a DTA < 0.5 across depression scores, with/without annuitized income.

Figure 4 depicts the moderating effect of annuitized income between having a debt-to-asset ratio of less than 0.5 given varying levels of self-reported health status. Individuals who have an annuitized income have a higher predicted probability of having a debt-to-asset ratio of less than 0.5 than those who do not have an annuitized income, at every level of health status. Similarly, the two lines are considerably parallel.

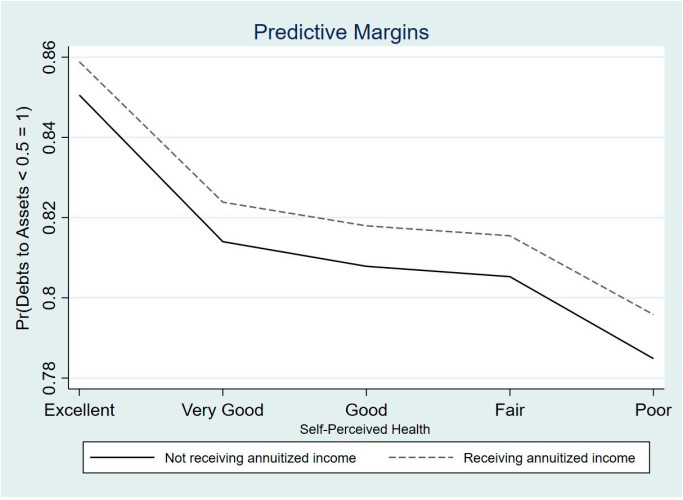

**Figure 4.** Predicted probability of having a DTA<0.5 across health conditions, with/without annuitized income.

## 4. Discussion

Focusing on a sample of older Americans, we conclude that holding a lifetime income stream will help people alleviate the stress of their financial situation and may marginally improve their financial capability. Although our findings do not support all of our hypotheses, they do point out the importance of measuring both subjective and objective financial well-being. We also see a similar result in the mediation effect of annuitized income helping those in poor mental and physical conditions to enhance their financial well-being. The positive effect of annuitized income is more significant on subjective financial well-being but not objective financial well-being for people with poor mental and physical health conditions. Although we found many interesting avenues of future research, we did not find any difference in the effect of annuitized income on financial well-being before and during the pandemic.

Our results demonstrate the importance of holding a stream of annuitized income to improve retirees' financial well-being, especially their subjective financial well-being. People are increasingly worried about running out of money during retirement, and the overall retirement satisfaction level is decreasing as retirement plans shift from DB to DC plans. Financial advisors should not only focus on helping clients accumulate enough retirement assets but also on helping them diversify their future retirement income streams. Our findings support pre-pandemic research, using 2017 data, that showed there is a positive association between subjective financial well-being and annuitized income from DB retirement plans (Lim and Lee 2021).

Our study also produces several results that are consistent with the existing literature. We do find a relationship between difficulty paying bills and debt, as individuals in our study who reported difficulty paying bills had lower odds of having a debt ratio of less than 0.5. This finding supports previous literature on the use of the debt-to-asset ratio as a measure of financial vulnerability (Lusardi et al. 2020). We also find that older Americans who identify as black have lower odds of having less debt than their white counterparts. Likewise, women had lower odds of having less debt than men. Several studies have found similar results on the association of race and gender since the 2008 recession in the U.S. (Castro Baker et al. 2019; Dymski et al. 2013; Lichtenstein and Weber 2015; Wyly and Ponder 2011).

There is also a need to keep financial advisors educated about the new lifetime income products for their DC plan clients. Lastly, although The Setting Every Community Up for Retirement Enhancement (SECURE) Act was passed in the U.S. House of Representatives in May 2019 and opened a door for retirement plan sponsors to introduce lifetime income products to DC plan participants, plan sponsors are still hesitant about taking action because of the fiduciary issue. Policymakers need to work on how to help plan sponsors feel comfortable bringing those "pension" elements back to their DC plan participants.

**Author Contributions:** Conceptualization, Q.S. and G.C.; methodology, Q.S.; software, Q.S. and G.C.; validation, Q.S. and G.C.; formal analysis, Q.S.; investigation, Q.S. and G.C.; data curation, G.C.; writing—original draft preparation, Q.S. and G.C.; writing—review and editing, Q.S. and G.C. All authors have read and agreed to the published version of the manuscript.

**Funding:** This research received no external funding.

**Data Availability Statement:** The data used in this study are from the University of Michigan's Health and Retirement Study. More information on this study can be found here: https://hrs.isr.umich.edu/. Accessed on 1 June 2022.

**Conflicts of Interest:** The authors declare no conflict of interest.

## Appendix A. Sensitivity Analysis

**Table A1.** Fixed-effects odds ratios for objective and subjective measures of financial well-being.

| | (1) Financial Strain | (2) Difficulty Paying Bills | (3) Debt-to-Asset < 0.5 | (4) Liquid Assets > Median HH Income |
|---|---|---|---|---|
| Dependent | | | | |
| Annuitized Income | 0.3944 * | 0.9599 | 0.9486 | 0.8713 |
| | (0.1554) | (0.2215) | (0.1938) | (0.1502) |
| Independent | | | | |
| Pandemic Year | 1.2578 | 0.2142 * | 1.1992 | 1.6497 |
| | (0.8851) | (0.1391) | (0.4017) | (0.7397) |
| Age | 0.7176 | 1.1403 | 1.0984 | 1.0843 |
| | (0.1274) | (0.1807) | (0.0950) | (0.1244) |
| Married | 2.3066 | 0.7633 | 0.7096 | 0.6187 |
| | (1.2717) | (0.2813) | (0.1781) | (0.2034) |
| Log of Annual Household Income | 1.0240 | 0.9828 | 0.9918 | 1.0956 *** |
| | (0.0279) | (0.0235) | (0.0154) | (0.0289) |
| Log of Non-Housing Net Worth | 0.9730 | 0.9799 | 1.0706 *** | 2.4295 *** |
| | (0.0209) | (0.0213) | (0.0143) | (0.1655) |

**Table A1.** *Cont.*

| | (1)<br>Financial Strain | (2)<br>Difficulty Paying Bills | (3)<br>Debt-to-Asset < 0.5 | (4)<br>Liquid Assets ><br>Median HH Income |
|---|---|---|---|---|
| Log of Housing Net Worth | 1.0220 | 0.9831 | 1.0264 * | 0.9911 |
| | (0.0234) | (0.0170) | (0.0108) | (0.0159) |
| Depression Score | 1.1135 | 1.1172 * | 0.9534 | 1.0457 |
| | (0.0650) | (0.0624) | (0.0300) | (0.0491) |
| Health | | | | |
| Reference Group is Excellent | | | | |
| Very Good | 0.4215 | 0.9386 | 0.8091 | 1.1321 |
| | (0.2008) | (0.3318) | (0.1969) | (0.2688) |
| Good | 0.5380 | 0.8441 | 0.8240 | 0.9098 |
| | (0.2659) | (0.3248) | (0.2175) | (0.2446) |
| Fair | 0.4634 | 0.9591 | 0.8290 | 0.6008 |
| | (0.2440) | (0.4129) | (0.2408) | (0.2057) |
| Poor | 0.9771 | 0.9808 | 0.8137 | 1.3168 |
| | (0.6648) | (0.5972) | (0.3130) | (0.7715) |
| *N* | 916 | 1152 | 2308 | 2884 |

Note: Observations from 2016 and 2020 HRS Data. *** Denotes statistical significance at the 1% level, and * denotes statistical significance at the 10% level.

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
