# Peer review of "Sustaining Retirement during Lockdown: Annuitized Income and Older American’s Financial Well-Being before and during the COVID-19 Pandemic"

_jrfm, doi:10.3390/jrfm16100432_

Round 1

Reviewer 1 Report

The research topic chosen by the authors is topical and the article could arouse interest from a potential audience. However, the proposed version of the article lacks one important section, a discussion of the results obtained and their interpretation in the context of other studies, see JRFM Instructions for Authors. The justification of the theoretical basis of the study and the methodology used should be deeper (only 12 references). 

Other recommendations: 

1) the necessary references are not always added when discussing and characterising the theoretical basis and methodology of the study, e.g. (a) describing the Cumulative Inequality Theory (CI theory) r. 65 – 75; (b) annuitized income effect on individuals' mental/physical health on financial well-being r. 87 – 96;

2) Debt to Asset Ratio< 0.5 is based on research by other authors, but does not assess the significance of the impact of this threshold on the study's findings;

3) when using abbreviations for the first time, they should always be deciphered, eg. IRA;

4) not all model parameters are explained, e.g. α1 , β1 γ1 δ1 υi and π2;

5) what is the rationale for believing that υi is constant for all models considered? 

6) odds ratios of 0.59 in line 222, 1.35 in line 252, 3.57 in line 260, 311 in line 262, 1.31 in line 267, 1.93 in line 272 differ slightly from the values in Table 2;

7) Fig.1 and 3 horizontal axis legend have typing errors;

8) References formatting should be in line with the requirements laid down by the JRFM.

Appropriate.

Reviewer 2 Report

Dear Authors,

The paper has been written well overall. I have some observation, though, about this paper.

1. Authors need to position their paper within the existing literature. This is where i believe this paper requires some improvement. That means, authors needs to cite more relevant papers all thorough the paper. They also need to add a section "Discussion" before the conclusion section to explain the implications of the findings. 

2. they need also to add some comparisons and contrasting between their findings and current reelvant literature.

3. Please add few more citations from the last 3 to 5 years sources/ references.

Thank you,

MM
